

# Importance of feature selection stability in the classifier evaluation on high-dimensional genetic data

Tomasz Łukaszuk and Jerzy Krawczuk

Faculty of Computer Science, Bialystok University of Technology, Bialystok, Poland

## ABSTRACT

Classifiers trained on high-dimensional data, such as genetic datasets, often encounter situations where the number of features exceeds the number of objects. In these cases, classifiers typically rely on a small subset of features. For a robust algorithm, this subset should remain relatively stable with minor changes in the training data, such as the replacement of a few samples. While the stability of feature selection is a common focus in studies of feature selection algorithms, it is less emphasized in classifier evaluation, where only metrics such as accuracy are commonly used. We investigate the importance of feature selection stability through an empirical study of four classifiers (logistic regression, support vector machine, convex and piecewise Linear, and Random Forest) on seven high dimensional, publicly available, gene datasets. We measure the stability of feature selection using Lustgarten, Nogueira and Jaccard Index measures. We employed our own cross-validation procedure that guarantees a difference of exactly $p$ objects between any two training sets which allows us to control the level of disturbance in the data. Our results show the existence of a drop in feature selection stability when we increase disturbance in the data in all 28 experiment configurations (seven datasets and four classifiers). The relationship is not linear, but resembles more of a hyperbolic pattern. In the evaluation of the tested classifiers, logistic regression demonstrated the highest stability. It was followed in order by support vector machine, convex and piecewise linear, with Random Forest exhibiting the lowest stability among them. This work provides evidence that all tested classifiers are very sensitive to even small changes in learning data in terms of features used by the model, while showing almost no sensitivity in terms of accuracy. The data and codes to reproduce the results in the article are available publicly on GitHub: https://github.com/tlukaszuk/feature-selection-stability-in-classifier-evaluation.

Corresponding author
Tomasz Łukaszuk,
t.lukaszuk@pb.edu.pl

## INTRODUCTION

In the field of bioinformatics, gene expression data analysis plays a vital role in understanding the mechanisms of gene regulation and identifying potential biomarkers for disease diagnosis and prognosis (*Gong et al., 2017*; *Keller et al., 2011*). With the advancements in high-throughput technologies, vast amounts of gene expression data are being generated, presenting new challenges and opportunities for data analysis. One of

the critical tasks in gene expression data analysis is the identification of the subset of genes that are most relevant to the phenotype of interest, which is often achieved through feature selection and classification techniques (*Deng, Xu & Wang, 2019*).

Feature selection involves the extraction of a subset of features from the original feature space to effectively reduce its dimensionality (*Liu & Motoda, 2012*). Notably, this process maintains the original feature space's structure intact, selecting only key features to reconstruct a low-dimensional feature space that preserves the same spatial relationships as the original.

Classification refers to the process of assigning predefined labels or classes to instances based on their characteristics or features. The goal is to develop a predictive model that can accurately classify new, unseen data into predefined categories (*Bishop, 2006*; *Duda, Hart & Stork, 2012*; *Fukunaga, 1972*).

While it is true that the accuracy of classifiers and the number of selected features are commonly reported in scientific publications on gene expression data analysis, the stability of feature selection, although equally important, is often overlooked.

Feature selection stability refers to the consistency of the selected features across random subsamples of the same dataset (*Khaire & Dhanalakshmi, 2022*). It is a critical aspect of feature selection as it reflects the generalizability and reproducibility of the selected features beyond the current dataset. Without assessing the feature selection stability, the selected features may not be robust enough to be used for downstream analysis or generalization to independent datasets, leading to unreliable and irreproducible results.

Therefore, it is essential to report the stability of feature selection along with the accuracy and the number of selected features in gene expression data analysis publications. This article presents a study of three metrics, using four publicly accessible high-dimensional datasets and evaluating four distinct classifiers that incorporate feature selection. Three of these classifiers utilize the $L_1$ norm—namely, logistic regression (LR), support vector machine (SVM), and our proprietary method that employs a convex and piecewise linear (CPL) criterion function. The fourth analyzed classifier is based on tree structures, specifically the Random Forest (RF) method.

Our article presents a special approach to cross-validation that allow us to better understand feature selection stability. In this approach we are able to control disturbance in data, by choosing a number of objects $p$ that are different between any two training datasets. This allows us to measure how the size of changes in datasets influences feature selection stability and other metrics for different models.

To summarize, in this article:

- we propose a methodology of generation of dataset subsamples for the use in feature selection stability evaluation;
- we empirically study the accuracy and feature selection stability in high-dimension gene expression data, as well as analyse seven gene datasets and four classification methods with embedded feature selection;
- we propose to combine feature selection stability with accuracy for classifier evaluation.

The rest of the article is organized as follows: in 'Background' we give a short background to our research related to the feature selection stability. Then, in 'Methods', we describe classifiers used, with special focus on our classifier, based on the CPL function, feature stability measures defined by Lustgarten, Nogueira and Jaccard, and also our cross-validation procedure called `trains-p-diff`. In 'Empirical Studies' we describe the datasets used and the experimental setup. In 'Results' we present the results of the experiments. In 'Discussion', we discuss the outcome we have achieved, particularly within the context of the most recent literature. Finally we conclude in 'Concluding Remarks'.

## BACKGROUND

In the last two decades, class prediction based on gene data has become a major topic in many medical fields such as oncology, and other diseases including multiple sclerosis (*Bomprezzi et al., 2003*). Such prediction refers to the categorical variable (most often binary) that can be *e.g.*, the presence or absence of a disease, a tumor sub-type or the response to a therapy (*Cruz & Wishart, 2006*; *Kourou et al., 2015*; *Pati, 2018*).

Usually in one study there is data available for less than 300 patients, where each patient is described by the thousands of gene expressions data. Such types of datasets with high dimension and low number of observations are very challenging for statistical methods, therefore many machine learning techniques are studied (*Boulesteix et al., 2008*).

Binary classification is the most basic task of machine learning and is often applied to gene data. Classifier models learn on high-dimensional datasets and, in addition, they automatically learn to use just a subset of features to make a prediction. Such subset selection by classifier is sometimes called intrinsic or embedded feature selection. Other types of feature selection, such as wrapper and filter methods (*Lazar et al., 2012*) were also studied to discover genes responsible for the outcome class, but they need to be used carefully because they could introduce the so called selection bias. This was outlined in our previous study (*Krawczuk & Łukaszuk, 2016*) and here we will focus on feature selection stability.

There has been much work done in the area of measuring the classification accuracy for gene expression data using error rate, sensitivity, specificity and even the area under receiver operating characteristic (ROC) curve (*Guo et al., 2014*; *Novianti et al., 2015*; *Roy et al., 2020*), but rarely were those measures combined with selection stability which may have fundamental practical meaning. If the model has low error but selects different genes when just one patient in the training set is replaced, it will not give insight on which genes could be the causes of disease and should be considered as unstable (*Al Hosni & Starkey, 2021*; *Khaire & Dhanalakshmi, 2022*).

Until recently, in the development of classification models, the focus was mainly on improving performance and prediction accuracy. Algorithms were often focused more on model complexity and minimising prediction errors (*Freitas, 2014*; *Galdi & Tagliaferri, 2018*). In addition, researchers and practitioners concentrated on aspects such as feature engineering, optimising hyperparameters or developing new machine learning algorithms (*Joy et al., 2016*; *Yang & Shami, 2020*). In recent years, however, an increased

awareness of the role of stability in feature selection and the awareness of the consequences of unstable feature selection on overall model performance can be observed (*Al Hosni & Starkey, 2021*; *Khaire & Dhanalakshmi, 2022*; *Yang et al., 2013*).

One of the key objectives in building predictive models is the ability to generalize. Classical approaches to generalization often focused on the overall ability of the model to deal effectively with new data and did not include an analysis of the stability of feature selection (*Kernbach & Staartjes, 2022*). The gradual understanding that the stability of feature selection is an important factor affecting a model's ability to generalize effectively led to a more detailed analysis of this issue in later years (*Al-Shalabi, 2022*; *Nogueira, Sechidis & Brown, 2018*; *Sechidis et al., 2020*).

The stability of feature selection algorithms was first studied by *Turney (1995)*. Currently, the topic of feature selection stability is a fairly important area of research in machine learning and predictive modelling. Researchers seek to understand how different feature selection methods affect the stability of models, as well as the practical implications of unstable feature selection on the overall performance of predictive models (*Al Hosni & Starkey, 2021*; *Huang, 2021*; *Piles et al., 2021*).

One of the earliest works proposing a framework for the stability of feature selection was (*Kalousis, Prados & Hilario, 2007*), which included, among others, three gene datasets. This study examined all three types of feature selection algorithms, including an embedded method, and also considered the aspect of classification quality in their research.

Another study by *Saeys, Abeel & Van de Peer (2008)* demonstrated that constructing ensemble feature selection techniques can enhance the stability of feature ranking and subset selection by using methods similar to those applied in supervised learning. They also emphasized the importance of evaluating the performance of the selected subset using classification metrics. Similar to our approach, they proposed the use of the F-measure—a well-known evaluation metric in data mining that represents the harmonic mean of stability (referred to as robustness in their work) and classification performance. In both *Saeys, Abeel & Van de Peer (2008)* and the previously mentioned work by *Kalousis, Prados & Hilario (2007)*, the Jaccard Index was used to measure stability.

## METHODS

This chapter outlines the methodologies and procedures employed in our study, providing a foundation for the ensuing analysis and discussion. We start with the examination of machine learned classifiers with the embedded features selection. This approach is beneficial in that many machine learning models can use a small part of the features available in training data which is particularly useful in high dimensional data. Then we describe in detail how the stability of selected features can be measured, which is the key aspect in bioinformatics and genetic data analysis where reproducibility and reliability are very important. This aspect of our research addresses the often overlooked, yet crucial, element of consistency in feature selections, which is essential for drawing reliable conclusions in high-dimensional data analysis. Lastly, we introduce our tailored cross-validation procedure we call `trains-p-diff`. This unique method is designed to ensure that exactly

$p$ objects differ between any two training sets—a strategy that allows us to control and measure the impact of dataset variability on feature selection stability and other metrics. This procedure not only enhances the rigor of our feature selection process but also provides deeper insights into the influence of data variability on model performance.

## Machine learning models with embedded feature selection

We focus only on these models, due to the following reasons:

- using any classifier on data such as gene expression, where the number of objects is much lower than the number of features will likely conduct feature selection anyway;
- we have been working on a classifier that minimizes convex and piecewise linear criterion function (CPL) with $L_1$ regularization which we would like to test;
- computational efficiency is a significant constraint, as employing wrapper methods such as SVM-RFE (*Guyon et al., 2002*; *Guyon & Elisseeff, 2003*) would render our extensive computations, which spanned weeks, unfeasible within a practical timeframe;
- and finally using other feature selection methods such as wrapper or filter would require the use of a classifier to measure accuracy anyway.

We can distinguish two types of machine learning models with embedded feature selection—one is based on $L_1$ regularization, while the other one is based on decision trees. One of the widely used anti-overfitting techniques is the regularization of the model parameters values by $L_1$ or $L_2$ norm. It can be used by many classifiers such as logistic regression, support vector machines, or neural networks. When both norms reduce overfitting, only $L_1$ norm tends to set weights for the features to zero, which works like feature selection. Let us introduce $L_1$ norm with our CPL classifier.

We assume that data is represented as $m$ feature vectors $\mathbf{x}_j = [x_{j1},...,x_{jn}]^T$ $(j = 1,...,m)$ of the same dimensionality $n$. The components $x_{ji}$ of the vectors $\mathbf{x}_j$ are called features. Each feature vector $\mathbf{x}_j$ is labelled with $y_j$ ($y_j \in \{-1,1\}$) its membership to the decision class. For example $y_j = -1$ describe patients suffering from a certain disease and the $y_j = 1$ describe patients without the disease. Then we define the convex and piecewise linear criterion function as:

$$\Phi_C(\mathbf{w},\theta) = \sum_{j=1}^{m} \max(0; 1 - y_j(\mathbf{w}^T\mathbf{x}_j + \theta)) + (1-C)\sum_{i=1}^{n}|w_i| \, , \tag{1}$$

where $\mathbf{w} = [w_1,...,w_n]^T \in \mathbf{R}^n$ is the weight vector, and $\theta \in \mathbf{R}^1$ is the threshold of the hyperplane $H(\mathbf{w},\theta)$ and $0 \geq C \geq 1$ is a hyperparameter that controls the strength of the regularization penalty. A lower value of $C$ increases the strength of the regularization and leads to more weights being pushed towards zero, resulting in a sparser model with fewer features. Conversely, a higher value of $C$ reduces the impact of regularization, allowing the model to fit the training data more closely.

The basis exchange algorithm allows us to find the minimum efficiently, even in the case of large multidimensional datasets (*Bobrowski, 1991*). In our experiments we used our own python implementation.

Similar to our criterion function $\Phi_C$ (Eq. (1)), $L_1$ regularization can be introduced to LR and SVM algorithms. In our experiments we used the scikit-learn (*Pedregosa et al., 2011*) implementation.

Another embedded feature selection method is one based on decision trees. In the process of building the tree on every internal node the feature needs to be selected for test that is based on gini or entropy criterion (*Tangirala, 2020*). For datasets with much more features than objects $n >> m$ only a small number of features will be used even for a fully grown tree without pruning. In practice many ensemble techniques are used with decision trees like bagging or boosting. We decided to use the RF algorithm in our experiments also from the scikit-learn package (*Pedregosa et al., 2011*).

**Feature selection stability measures**

The issue of feature selection method sensitivity to minor changes in training data has been explored by several authors. If the selection of features changes significantly when a different sample from the same training data is used, the method is deemed unstable. On the other hand, if the feature subset remains largely unchanged despite alterations in the data, the method is considered stable. Although this notion is straightforward, there is currently no universally accepted metric for quantifying stability, and various suggestions have been put forth in the literature.

One of the first measures used to assess stability were those based on the intersection of feature sets, such as the Tanimoto distance (*Kalousis, Prados & Hilario, 2007*) and the Jaccard Index (*Saeys, Abeel & Van de Peer, 2008*). These measures allow for the evaluation of the similarity between feature sets of different cardinalities but do not account for the total number of features, and therefore, do not provide a correction for chance. However, in our case, where the total number of features is much higher than the number of selected features, this correction is minimal.

$$S_J(F^{c_i}, F^{c_j}) = \frac{|F^{c_i} \cap F^{c_j}|}{|F^{c_i} \cup F^{c_j}|} = \frac{r}{c_i + c_j - r} \qquad (2)$$

where $F^{c_i}$ and $F^{c_j}$ are two subsets of features with cardinalities $c_i$ and $c_j$, respectively, and $r$ is the number of shared features. This measure ranges from 0 to 1, taking the value of 0 when there are no common features and the value of 1 when both subsets are identical.

In our previous research, we used a measure proposed by *Lustgarten, Gopalakrishnan & Visweswaran (2009)* as an extension of the Kuncheva similarity measure (*Kuncheva, 2007*). The Kuncheva metric introduces a correction for chance but is applicable only to two datasets with the same cardinality. Lustgarten's extension allows for its use with subsets $F^{c_i}$ and $F^{c_j}$ of different cardinalities $c_i$ and $c_j$, which is the usual case in embedded feature selection methods. It is calculated as follows:

$$S_L(F^{c_i}, F^{c_j}) = \frac{r - \frac{c_i c_j}{n}}{min(c_i, c_j) - max(0, c_i + c_j - n)}, \qquad (3)$$

where $n$ is the total number of features. This measure ranges from $(-1, 1]$ and provides a correction for chance. Negative values correspond to situations where common features

appear less frequently than the expected overlap under random selection. Feature stability is then defined as the average similarity of all pairs of feature sets:

$$ASM = \frac{2}{k(k-1)} \sum_{i=1}^{k-1} \sum_{j=i+1}^{k} S_{J/L}(F^{c_i}, F^{c_j}) , \tag{4}$$

where $k$ is the number of obtained features sets.

Lately *Nogueira, Sechidis & Brown (2018)* proposed a new feature selection stability measure based on the frequency of selected features.

$$Z = \begin{bmatrix} z_{11} & z_{12} & z_{13} & \dots & z_{1n} \\ z_{21} & z_{22} & z_{23} & \dots & z_{2n} \\ \dots \\ z_{k1} & z_{k2} & z_{k3} & \dots & z_{kn} \end{bmatrix} \tag{5}$$

Each row $Z_k$ of $Z$ matrix refers to one feature selection subset, columns represents each feature, then observed frequency of selection of $f$ feature is $p_f = \frac{1}{k}\sum_{i=1}^{k} z_{if}$. While it is still a generalization of Kuncheva measure in the sense that for subsets with the same cardinality it has the same values, it is computationally effective and holds the so called Maximum Stability (Deterministic Selection) property. The Nogueira features stability measure $\phi(Z)$ reaches its maximum, if-and-only-if all feature sets in $Z$ are identical. *Nogueira, Sechidis & Brown (2018)* show that their measure holds this property while the Lustgraten measures violate the backward implication of it.

$$\phi(Z) = 1 - \frac{\frac{1}{n}\sum_{f=1}^{n} s_f^2}{\frac{\bar{n}}{n}\left(1 - \frac{\bar{n}}{n}\right)} \tag{6}$$

where $s_f^2 = \frac{k}{k-1} p_f (1 - p_f)$ is the unbiased sample variance of the selection of the $f$ feature and $\bar{n} = \frac{1}{k}\sum_{i=1}^{k}\sum_{f=1}^{n} z_{if}$.

It it worth noticing that both measures hold the other four desired properties mentioned by *Nogueira, Sechidis & Brown (2018)*:

1. Fully defined. The stability estimator $\phi(Z)$ (Eq. (6)) should be defined for any collection $Z$ of feature sets, thus allowing for the total number of features selected to vary.
2. Strict monotonicity. The stability estimator $\phi(Z)$ (Eq. (6)) should be a strictly decreasing function of the sample variances $s_f^2$ of the variables $Z_f$ (column in the matrix $Z$).
3. Bounds. The stability $\phi(Z)$ (Eq. (6)) should be upper/lower bounded by constants not dependent on the overall number of features or the number of features selected.
4. Correction for chance. Subsets of features can share some of the features even if selected randomly.

### `trains-p-diff` procedure for generating dataset splits

Cross-validation is a technique used in machine learning to evaluate the performance of a model by splitting the dataset into training and testing sets. The basic idea is to train the model on a portion of the data and then use the remaining data to evaluate the model's performance. Typically, we divide a set of objects into a fixed number of $k$ roughly equal

**Table 1 Example demonstrating the operation of the `trains-p-diff` procedure.** Assumptions: the dataset $D$ consists of $|D| = 10$ objects, the training sets in each split are to be of size $t = 6$, the training set from any split is to have $p = 2$ objects not present in the training set from the other split, i.e., each pair of training sets from different splits is to have $t - p = 6 - 2 = 4$ objects in common. The training and test sets are presented as object indexes in the range 0–9.

| Split | Training set | Test set |
|---|---|---|
| 1 | [1, 3, 4, 6, 7, 9] | [0, 2, 5, 8] |
| 2 | [0, 1, 3, 5, 7, 9] | [2, 4, 6, 8] |
| 3 | [0, 1, 4, 5, 6, 9] | [2, 3, 7, 8] |
| 4 | [0, 3, 4, 5, 6, 7] | [1, 2, 8, 9] |

portions, and then create a training set from $k - 1$ portions and, using the remaining one portion, assess the quality of the model. The procedure is repeated $k$ times, and each time a different portion of the data acts as the test set.

In our research, the aim is to evaluate the discriminant model in terms of the stability of the features it uses and the impact of the composition of the training set on the features selected by the model. With this aim in mind, it is particularly important to construct the training sets correctly. In each cross-validation step, the number of objects in the training set should be the same and each training set should differ from any other training set used in the other cross-validation steps by a precisely specified number $p$ ($p \in \{1, 2, \ldots\}$) objects. In order to meet these conditions, we have developed our own method for generating training and test sets for cross-validation.

The parameters of the `trains-p-diff` procedure are the size of the training set $t$ in each split generated in successive cross-validation steps and the number of objects $p$ by which the training sets of any two splits differ. The dataset $D$ on which the splits are to be made should have a count of at least $t + p$ ($|D| \geq t + p$). The procedure will result in train/test splits with counts of $t$ and $|D| - t$ respectively. Each selected pair of training parts from any two splits will have exactly $t - p$ common objects and $p$ objects occurring in only one of the sets from the pair. An example showing how the procedure works is shown in Table 1.

Using the `trains-p-diff` procedure described above, we can obtain reproducible training and test sets that allow the results of different discriminant models to be compared. We can also accurately control the size and degree of differentiation in the composition of the training sets, allowing us to assess the stability of the model.

## EMPIRICAL STUDIES

In this chapter we present a detailed overview of our experimental studies, encompassing the datasets used, the setup of our experiments, and a comprehensive presentation of the results obtained. We start with the description of the datasets that have been chosen for their relevance and potential to provide insightful revelations in the realm of high-dimensional genetic data analysis. We outline the characteristics of each dataset in terms of the number of objects and features. In the next section, we provide clear and replicable framework of our experiments, ensuring that our experiments can be validated and reproduced. Finally, we present and interpret our empirical results.

**Table 2  Genetic datasets used.**

| Dataset | #objects | #features | Class sizes |
|---|---|---|---|
| Breast | 289 | 35,981 | 143/146 |
| Colorectal | 194 | 49,386 | 97/97 |
| Leukemia | 101 | 54,675 | 60/41 |
| Liver | 165 | 47,322 | 115/50 |
| Prostate | 124 | 12,620 | 64/60 |
| Renal | 143 | 54,675 | 71/72 |
| Throat | 103 | 54,675 | 74/29 |

## Datasets

We selected seven gene expression datasets named: *Breast, Colorectal, Leukemia, Liver, Prostate, Renal* and *Throat*. They were obtained from the Curated Microarray Database (CuMiDa) repository provided at https://sbcb.inf.ufrgs.br/cumida (*Feltes et al., 2019*). All datasets were characterized by a significant number of features, in this case describing the expression of specific genes, and the presence of two decision classes. Table 2 shows the most important parameters of the datasets.

The CuMiDa database is a collection of several dozen carefully selected microarray data sets for *Homo sapiens*. These data sets were chosen from over 30,000 microarray experiments available in the Gene Expression Omnibus (GEO) (*Edgar, Domrachev & Lash, 2002*), following strict filtering and evaluation criteria. In addition to technical aspects, proper data curation of GEO datasets also supports ethical standards. Thorough documentation of experimental methods, participant consent, and compliance with ethical guidelines help protect the rights and privacy of individuals contributing to scientific research.

## Experimental setup

For every experiment we used the `trains-p-diff` procedure described earlier for $p \in P$ ($P = \{1, 2, 3..10, 12, 14, \ldots, 20\}$), and for every $p$ there was usually 10 of datasets splits to train and test, sometimes less if the dataset was small and the $p$ high. This was repeated three times for different random seeds. For every split a classifier was trained and metrics were obtained. Accuracy was calculated on a test set, number of selected features and feature selection stability metrics were calculated based on a fitted classifier model (see Fig. 1).

We performed two kinds of experiments - in the first one, called the single classifier, we ran one classifier on all datasets with different values for $C$. From this experiment we can see how different values of $C$ influence the results. This experiment did not really allow us to compare different classifiers, which is why we performed the second experiment where for each dataset we ran all classifiers with the $C$ parameter set in such a way, that all of them selected a similar number of features (see Fig. 2). In this design we can compare classifiers and for example say which of them is more stable in terms of selecting the same features.

As mentioned earlier, implementations of classifiers LR, SVM and RF offered by the scikit-learn (*Pedregosa et al., 2011*) library, as well as our own implementation of the CPL classifier, were used in the experiments. Most of the parameters of the classifiers were left at default values apart from those listed below:

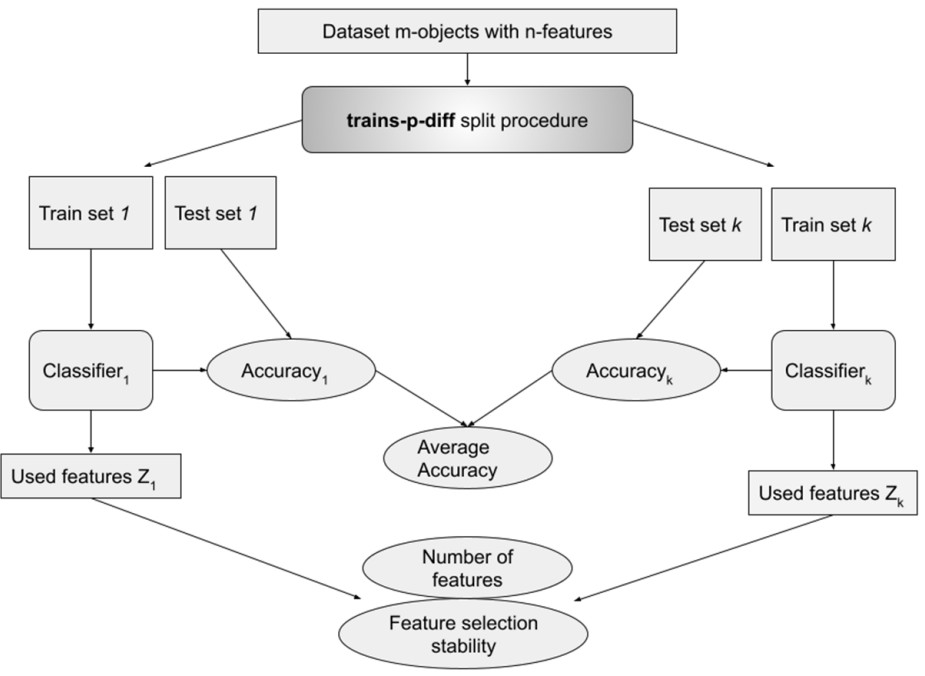

**Figure 1 Metrics calculation with `trains-p-diff` procedure.** This procedure was repeated three times for each tested value of $p$ between 1 and 20. Dataset was typically split $k = 10$ times, although fewer splits were used for smaller datasets with high values of $p$. A classifier was trained for each split, and relevant metrics were obtained.

```
LR: LogisticRegression(C, penalty ='l1', solver ='liblinear', max_iter
    =100000)
SVM: LinearSVC(C, penalty ='l1', dual =False, max_iter =100000)
RF: RandomForestClassifier(n_estimators =100, random_state =0,
    min_samples_leaf =2)
CPL: GenetClassifier(C)
```

In the case of the LR, SVM and CPL classifiers, the influence on feature selection, in particular on the number of selected features, was obtained by manipulating the values of the $C$ parameter. In the RF classifier, on the other hand, features were selected on the basis of a threshold value set for the collection of `feature_importances_` (the Gini importance) created on the basis of the trained model.

## RESULTS

The results for the 'single classifier' experiment are displayed in Figs. 3 and 4, where the $X$-axis consistently represents $p$ - the number of differing objects between any two training sets. We anticipate minimal changes in the selected features for a small $p$ (indicating slight differences in the training set), but as $p$ increases, these changes may also grow. This trend is evident in the first three columns, which assess feature selection stability using

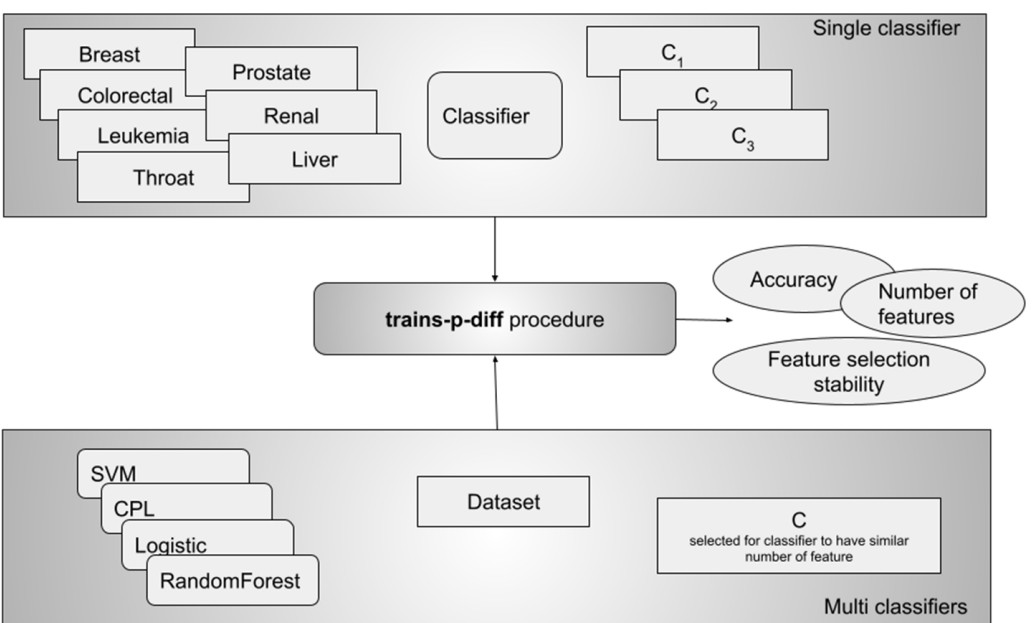

**Figure 2 The experimental design included two approaches: testing a single classifier with various values across all datasets, and using a single *C* value per classifier to achieve a similar number of selected features (multi classifiers).** The second approach allowed us to compare results between classifiers.

Lustgarten, Nogueira and Jaccard Index measures. The decrease in stability doesn't appear linear but more hyperbolic in nature. We observe better stability at lower values of the regularization parameter *C* (indicating model complexity), meaning that models select fewer features, as shown in the fourth column's charts. These charts also reveal that the number of selected features is independent of the randomness of the training set (or the value of *p*) and depends solely on *C*. The last column presents the accuracy measured on the test set, which, again, seems unaffected by the value of *p* and is influenced by *C* in certain instances. This collection of charts distinctly showcases the different behaviors of the RF classifier compared to other classifiers employing $L_1$ regularization.

The outcomes of the 'multi classifiers' experiment are shown in Fig. 5. We regulated each classifier to select a comparable number of features: 60 for the *Breast* dataset, 20 for *Colorectal*, *Leukemia* and *Liver* datasets and 25 for *Prostate*, *Renal* and *Throat* datasets. The exact feature counts are shown in the fourth column, with the remaining columns mirroring those from the initial experiment involving single classifiers. In this scenario, each line represents a different classifier, facilitating direct comparisons. The first three columns of the chart, which illustrate feature selection stability, clearly reveal that RF performs the poorest, while the others are more closely matched. However, they can be ranked, starting with LR as the most effective, followed by SVM, and then CPL, as detailed in Table 3. Notably, LR exhibits unique behavior in the *Throat* dataset, demonstrating high feature stability across all *p* values but simultaneously the lowest test set accuracy among all classifiers, as seen in the fourth column. Regarding accuracy, there is no definitive best performer; RF lags behind in the *Breast* dataset, while CPL and SVM excel in the *Throat*

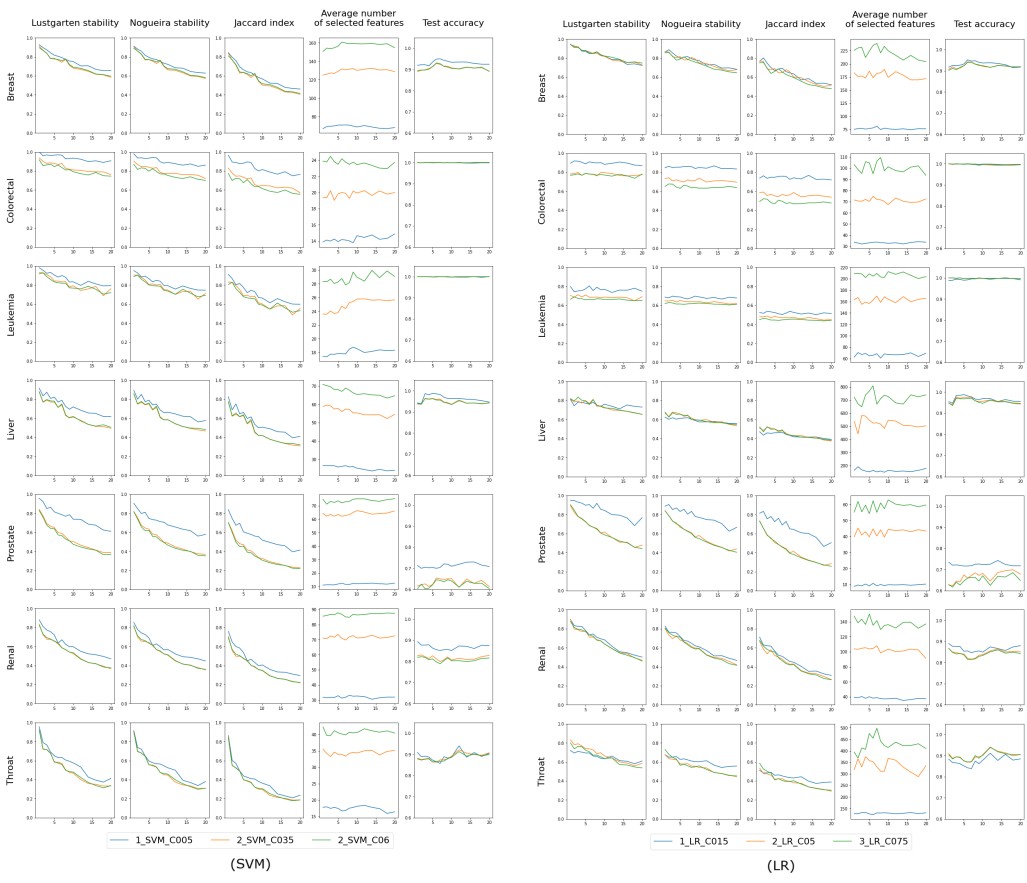

**Figure 3  Results for classifiers SVM and LR, each tested with three varying complexity parameter values (indicated by colors).** The first three columns in the chart display the stability of feature selection as measured by Lustgarten, Nogueira and Jaccard Index metrics. The fourth column presents number of selected features, and the last column indicates the test accuracy. The $X$-axis represents the values of $p$ used in `trains-p-diff` cross-validation procedure, *i.e.,* the number of objects by which any two training splits differ.

dataset. For the *Colorectal* and *Leukemia* datasets, the accuracy for all classifiers is very close to 1.0. Therefore, in these cases, examining the stability of the selected features becomes more important.

To enable a direct comparison of classifiers, we have computed the average values for feature selection stability $\overline{ASM}$ (Eq. (7)), as measured by the Lustgarten metric, and displayed them in Table 3. In addition, we suggest employing the harmonic mean of feature selection stability $\overline{ASM}$ and test accuracy $\overline{acc\_test}$ (see Eq. (8)) to derive a singular measure that accounts for both aspects of classifier quality. The findings are documented in Table 4. Given that the accuracy was relatively consistent across classifiers, the results closely mirror the feature selection stability alone, indicating that the LR model performs the best, followed by SVM and CPL, with RF ranking the lowest. It is also worth noting that the $F_\beta$ score can be adjusted with different $\beta$ values to place greater emphasis on either

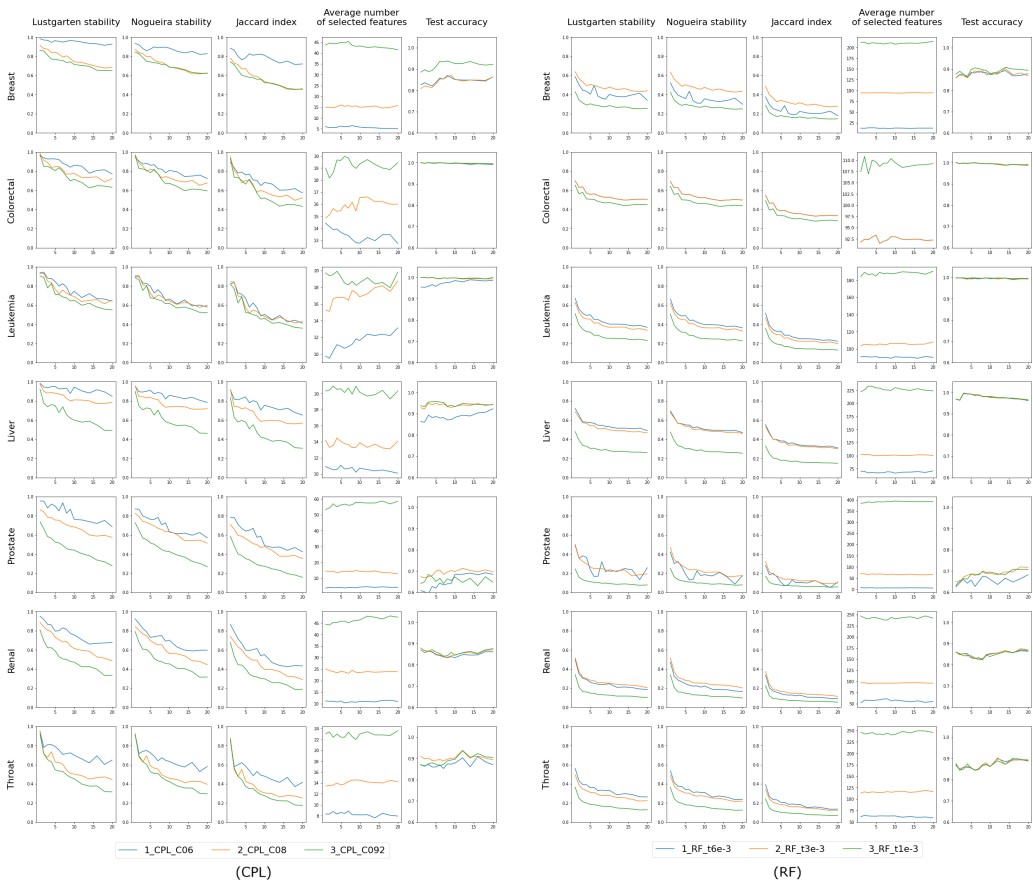

**Figure 4 Results for classifiers CPL and RF, each tested with three varying complexity parameter values (indicated by colors).** The first three columns in the chart display the stability of feature selection as measured by Lustgarten, Nogueira and Jaccard Index metrics. The fourth column presents number of selected features, and the last column indicates the test accuracy. The X-axis represents the values of $p$ used in `trains-p-diff` cross-validation procedure, *i.e.,* the number of objects by which any two training splits differ.

stability or accuracy.

$$\overline{ASM} = \frac{\sum_{p \in P} ASM_p}{|P|} \tag{7}$$

where $ASM_p$ is $ASM$ (Eq. (4)) stability measure value obtained in `trains-p-diff` procedure with fixed value of $p$.

$$F_1 = \frac{2}{\frac{1}{\overline{ASM}} + \frac{1}{\overline{acc\_test}}} \tag{8}$$

where $\overline{ASM}$ and $\overline{acc\_test}$ indicate the average feature stability (Eq. (7)) and average test classification accuracy, respectively. Averaging is done over all tested values of the $p$ parameter in `trains-p-diff` cross-validation procedure.

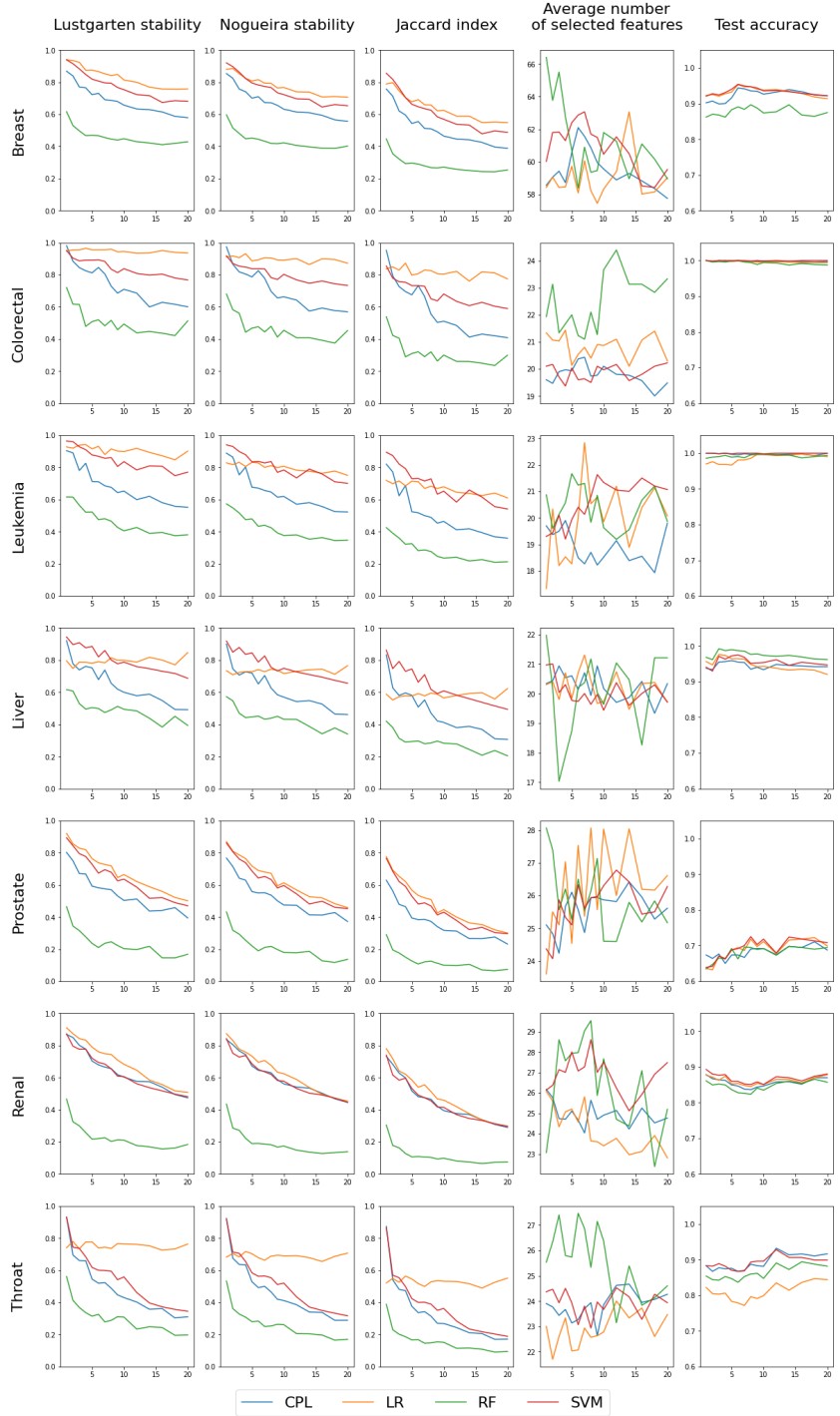

**Figure 5  Results for four classifiers, distinguished by color, set to select a comparable number of features (shown in the fourth column) by selecting the complexity parameter value.** The first three columns in the chart display the stability of feature selection as measured by Lustgarten, Nogueira and Jaccard Index metrics. The forth column presents number of selected features, and the last column indicates the test accuracy. The $X$-axis represents the values of $p$ used in trains-p-diff cross-validation procedure, *i.e.,* the number of objects by which any two training splits differ.

**Table 3** **The average values for feature selection stability $\overline{ASM}$ (Eq. (7)), as measured by the Lustgarten metric obtained by the tested classifiers for the datasets used.** Each number in the table is the appropriate, for a particular classifier and dataset, averaged over the considered values of $p$, the value of the Lustgarden stability measure obtained in the 'multi classifiers' experiment (see Fig. 5, the first column).

|      | Breast | Colorectal | Leukemia | Liver | Prostate | Renal | Throat |
|------|--------|------------|----------|-------|----------|-------|--------|
| LR   | 0.841  | 0.949      | 0.907    | 0.793 | 0.698    | 0.713 | 0.752  |
| SVM  | 0.786  | 0.850      | 0.853    | 0.813 | 0.662    | 0.652 | 0.569  |
| CPL  | 0.697  | 0.750      | 0.693    | 0.663 | 0.566    | 0.659 | 0.508  |
| RF   | 0.463  | 0.511      | 0.471    | 0.491 | 0.241    | 0.232 | 0.308  |

**Table 4** **Harmonic means (Eq. (8)) of the Lustgarten stability measure and test accuracy obtained by the tested classifiers for the datasets used.** Each number in the table is the harmonic mean of the Lustgarden stability measures averaged over the considered values of $p$ and the test accuracy averaged over the considered values of $p$, for a particular classifier and dataset, in the 'multi classifiers' experiment (see Fig. 5, the first and the last columns).

|      | Breast | Colorectal | Leukemia | Liver | Prostate | Renal | Throat |
|------|--------|------------|----------|-------|----------|-------|--------|
| LR   | 0.884  | 0.972      | 0.944    | 0.864 | 0.693    | 0.781 | 0.779  |
| SVM  | 0.854  | 0.919      | 0.921    | 0.879 | 0.677    | 0.745 | 0.694  |
| CPL  | 0.794  | 0.857      | 0.818    | 0.779 | 0.618    | 0.744 | 0.647  |
| RF   | 0.606  | 0.675      | 0.638    | 0.654 | 0.357    | 0.364 | 0.453  |

## DISCUSSION

In the discussion section three matters need to be pointed out: firstly, the uniqueness of our `train-p-diff` procedure and how other researchers address perturbaion in data in the feature selection stability context; secondly, our focus on comparing feature sets with different cardinality opposed to top-k features from ranking; and finally, the discussion of our results and future work.

To the best of our knowledge, there have been no published studies that have evaluated the stability of feature selection algorithms using the specific approach applied by us—by rigorously controlling perturbation in training sets through defining the exact number of differing objects. Nevertheless, we will refer to some articles that performed experiments with the described levels of perturbations in the data.

In the latest study by *Barbieri, Grisci & Dorn (2024)*, two levels of perturbation in the training data—small and significant—were tested to analyze their effect on the stability of feature selection. The small perturbation was achieved by sampling 90% of the instances with reposition. The significant perturbation was obtained through a bootstrap-like sampling with repetition, where an average of 63.2% of the original samples were retained. In our study, both perturbations are considered rather significant, as it was observed that altering even a single instance in the training set can lead to different feature selections.

The work, cited in the previous paragraph, by *Barbieri, Grisci & Dorn (2024)* is closely related to our research, particularly since some of the datasets were sourced from CuMiDa (*Feltes et al., 2019*), the same repository we used. For example, on the *Liver* dataset, they report an F-measure between 0.9 and 1.0 for SVM and RF, with the number of selected features ranging from five to 200, which is consistent with our experimental results. While we use accuracy as our metric, they stated, ''Because datasets are very balanced, the accuracy

 

metric yields results similar to the F-measure''. For the *Prostate* dataset, both their results and ours were lower, around 0.7. In terms of stability, they used the Kuncheva metric, which can only be applied to two feature subsets of the same cardinality. This approach was suitable for their experimental framework, so precise results cannot be directly compared with ours. However, our work can be seen as an extension of theirs, as we explore different levels of data perturbation and integrate feature selection with model building.

Three levels of training data perturbations were studied by *Pes (2020)*. Perturbation in her experiments were not fully controlled as in our case, but were based on randomness. Specifically 0.9 fraction of input dataset were repeatedly randomly sampled (without replacement) to create different training sets. This procedure created training datasetets which could differ in the range between 0 and 10%. Experiments were also performed using 0.8 and 0.7. Interestingly, the drop in stability in these results between 0.9 and 0.8 is much higher than beetween 0.8 and 0.7, which is exactly what was observed in our experiments for growing disturbancy in the data (hyperbolic shape).

In another deep study on stability of feature selection by *Kalousis, Prados & Hilario (2007)* ten-fold cross-validation was used where the overlap of training instances among the different training folds was reported to be 78%. Different amounts of perturbation in data were not considered whatsoever, but their size was directly noticed.

Secondly, we would like to discuss how we compared selected features. In our experiments, the model is allowed to choose the appropriate number of features based on the data presented during training, so the number of features in each run was usually different. In contrast, some other techniques select the top-K features, resulting in identical cardinalities. For example, in the articles mentioned in the previous paragraphs, stability was measured on the top five, 10, 20, *etc.*, best features using the Kuncheva metric (*Kuncheva, 2007*).

We previously worked with the Lustgarten extension for the Kuncheva metric (*Lustgarten, Gopalakrishnan & Visweswaran, 2009*), but recently, we discovered an interesting proposal from *Nogueira, Sechidis & Brown (2018)*. The Nogueira metric has a strong theoretical foundation, although it is not widely used. Our experiments show only a small difference between these two metrics, and they generally lead to the same conclusions. Many other studies in this area also use metrics based on the intersection of feature sets, such as the Jaccard Index (*Saeys, Abeel & Van de Peer, 2008*), so we included it in our results as well.

Finally we would like to comment on our research, which we think can be seen as a proposal of a new dimension on which feature selection stability can be studied - precisely defined perturbation in training data. The most common dimension found in literature is the top $k$ number of features, after—of course—comparing different algorithms. For example, in *Kalousis, Prados & Hilario (2007)* stability metrics for 10 top features were reported. Less often some hyperparameters of the algorithms are tested. When we decided to measure stability in function of perturbation in data we were faced with the challanges of eliminating other factors that may influence it—namely, those other dimensions. This is why we performed two types of experiments. In the first, together with perturbation in data we also checked the regularization parameter $C$ (hyperparameter), which showed us

how stability depends on both perturbation and regularizaton. The second experiment was different in the sense that we wanted to fix everything for all algorithms in a similar way to be able to compare them. We decided to fix the number of features the algorithms selected by choosing the specific value for regularization. We were aware that those were not all possible ways of studying feature selection stability with controlled perturbation in the data, but those two experiments gave us some valuable insights.

The first experiment showed us that the relation between more stability and more regularization (less features) is not as obvious as it was expected, since it could be argued that if less features were selected, they should be only strong, identical features. In our experiments this could be observed only for some combinations of classifiers and datasets.

The second experiment was designed by us in order to be able to compare classifiers directly. It clearly shows the order for the regularization based classifiers, with the LR being most stable, followed by SVM and our CPL. It can be argued that SVM should be stable in small perturbations in data since the hyperplane is supported just by a few instances from the data (no more than the number of selected features). If we do not remove those instances from the training set it should stay the same. LR stability was a real surprise for us, but it can be noticed that this basic benchmarking method performs very well in many cases. Lower stability of CPL was a slight disappointment for us, because it was hoped that stability could be the strongest property among the two direct competitors - LR and SVM. It was known from previous experiments that they are similar in terms of accuracy (*Krawczuk & Łukaszuk, 2016*).

The CPL stability in comparison to others was underwhelming. We can see how this method can be developed towards improvement. *Bobrowski (1996)* proposed constructing a CPL function based on so-called dipols (two instances from the dataset). Dipols could be clear (two objects from the same class) or mixed (two objects from two different classes). The idea is to look for the hyperplane that will go through as many mixed dipoles as possible. The number of dipols is the squared number of instances, and although computationally it will be more expensive, it can provide more stability for the selected features.

Another direction we would like to explore is to investigate more precisely how the regularization parameter $C$ influences the results of the CPL classifier, similar to the approach in *Nogueira & Brown (2016)*, where both feature selection stability and accuracy were studied over a wide range of regularization parameters for the LR classifier. We aim to employ a specific method of selecting $C$ that minimizes the reduction of one feature, as proposed in *Bobrowski & Łukaszuk (2011)*.

To conclude the discussion, it is important to address the limitations of the presented research. Firstly, we relied solely on data from a single domain, gene expression, made available by one data repository. These data may have unique characteristics, so in future research, it would be beneficial to use data from other domains as well, such as representations of text documents, which also have a much larger number of features than objects. The second limitation worth noting is related to one of the computational procedures implemented during the experiments. This concerns the choice of the value of the C parameter in the case of LR, SVM, and CPL classifiers, or the cut-off threshold in the case of the RF classifier, both of which control the complexity of the models. Determining

the appropriate value of these parameters, which varies for each classifier and dataset, in order to select the desired number of features, often required numerous time-consuming trials. In the future, automating this process as much as possible would allow for more efficient execution of experiments.

## CONCLUDING REMARKS

Our research examines four classifiers with built-in feature selection capabilities - three utilizing $L_1$ regularization and one based on decision trees. We conducted a series of experiments on seven high-dimensional gene expression datasets, using three metrics to assess feature selection stability. Alongside the stability, we always evaluate accuracy, as high stability paired with low accuracy does not sufficiently differentiate one classifier from another. In our studies, we employed a novel cross-validation approach named `trains-p-diff`, ensuring that each pair of training sets differs by exactly $p$ items. This method allowed us to assess feature selection stability at various levels of data disturbance.

Minor modifications to the dataset, such as replacing a single item, can lead to a different subset of features being chosen by classifiers, with stability measures dropping below 1. As the level of disturbance escalates, stability generally decreases. This decline is initially more marked but lessens as the disparity in the data (*i.e.,* the value of $p$) increases, exhibiting a hyperbolic curve-like pattern. This trend is consistent across all classifier types, datasets, and stability measures. However, the extent of stability does vary among different classifiers and datasets. In our experiments, the RF classifier consistently showed the lowest stability, while the LR, based on $L_1$ regularization, demonstrated the highest stability. Interestingly, the accuracy of the classifiers seemed largely unaffected by data disturbances, with most classifiers performing comparably well. However, the RF performed least effectively on the *Breast* dataset, and the LR on the *Throat* dataset.

In summary, feature selection stability is a crucial aspect of classifier evaluation in gene expression data analysis. It aids the identification of a robust and informative subset of genes, enhances the interpretability of results, and contributes to the generalization and reliability of classification models in high-dimensional biological data contexts.

In this article, we did not focus on methods to improve stability or investigate factors influencing it, but merely measured this aspect of feature selection across several datasets and classifiers. In the future, it would be valuable to examine what affects this stability and how it can be enhanced.

### Funding

This work was supported by the Bialystok University of Technology (No. WZ/WI-IIT/4/2023). The funders had no role in study design, data collection and analysis, decision to publish, or preparation of the manuscript.

### Grant Disclosures

The following grant information was disclosed by the authors:

Bialystok University of Technology: WZ/WI-IIT/4/2023.

## Competing Interests

The authors declare there are no competing interests.

## Author Contributions

- Tomasz Łukaszuk conceived and designed the experiments, performed the experiments, analyzed the data, prepared figures and/or tables, authored or reviewed drafts of the article, and approved the final draft.
- Jerzy Krawczuk conceived and designed the experiments, performed the experiments, analyzed the data, prepared figures and/or tables, authored or reviewed drafts of the article, and approved the final draft.

## Data Availability

The data and codes are available at GitHub and Zenodo:

- https://github.com/tlukaszuk/feature-selection-stability-in-classifier-evaluation.

- tlukaszuk. (2024). tlukaszuk/feature-selection-stability-in-classifier-evaluation: v1.0.0 (v1.0.0). Zenodo. https://doi.org/10.5281/zenodo.13936935.

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
