# Peer review of "Importance of feature selection stability in the classifier evaluation on high-dimensional genetic data"

_PeerJ, doi:10.7717/peerj.18405_

## Round 0.1 · original submission · Major Revisions

Please check reviewers' comments to improve the quality of your submission.

Reviewer 1 ·

Basic reporting

This work studies the feature selection stability of four classifiers on four gene datasets.

The paper is well presented and is easy to read. However, it requires some work to improve its quality:
The state of the art must be extended. There are many works in which stability has been included in feature selection papers. Some examples of works that shoul be included:
1. Nogueira, S., & Brown, G. (2016). Measuring the stability of feature selection. In Machine Learning and Knowledge Discovery in Databases: European Conference, ECML PKDD 2016, Riva del Garda, Italy, September 19-23, 2016, Proceedings, Part II 16 (pp. 442-457). Springer International Publishing.
2. Kalousis, A., Prados, J., & Hilario, M. (2007). Stability of feature selection algorithms: a study on high-dimensional spaces. Knowledge and information systems, 12, 95-116.
3. etc.

Experimental design

- The method introduced is not clear after reading it description. I would recommend to introduce a Figure or example to improve its readability.
- I recommend to include the Jaccard index (the pairwise averaging of the Jaccard index) as a stability measure
see Saeys, Y., Abeel, T., & Van de Peer, Y. (2008). Robust feature selection using ensemble feature selection techniques. In Machine Learning and Knowledge Discovery in Databases: European Conference, ECML PKDD 2008, Antwerp, Belgium, September 15-19, 2008, Proceedings, Part II 19 (pp. 313-325). Springer Berlin Heidelberg.
- Why have you selected only four gene datasets? The experiments must include more gene datasets.

Validity of the findings

- The authors claim in the discussion (see lines 336-339) that it is not common that stability of feature selection is measured on feature sets of different cardinality. This is not true. There are many other works that calculate stability between feature subsets of different cardinality using the Jaccard index. This is why the authors must improve the state of the art section.
- In order to improe the discussion and conclusion sections, more datasets must be included in the study.
-

Additional comments

No additional comments.

Reviewer 2 ·

Basic reporting

The article is written in clear and professional English, with no major issues in language or grammar that might hinder understanding.
The literature review is thorough, providing a solid background and context for the study. The references are up-to-date and relevant to the field.
The structure of the article is logical, with sections clearly delineated. Figures and tables are well-designed and enhance the comprehension of the data.
Raw data is provided, which supports the transparency and reproducibility of the research.
The paper is self-contained, presenting all necessary information to understand the study and its findings without requiring additional resources.

Experimental design

The research is original and fits well within the Aims and Scope of the journal, contributing new insights to the field.
The research question is well-defined, relevant, and addresses a meaningful gap in the existing literature.
The study is conducted rigorously, adhering to high standards of technical and ethical practice.
The methods are described in detail, with enough information provided to allow other researchers to replicate the study. However, a few areas could benefit from further clarification.

Validity of the findings

The findings are robust and supported by statistically sound analyses. The data is well-controlled, ensuring the validity of the results.
The conclusions are well-stated and directly linked to the original research question, providing clear answers based on the data presented.
The study’s impact and novelty are noteworthy, offering valuable contributions to the literature.
The rationale for the study is clearly explained, and the potential benefits to the field are well-articulated.

Additional comments

Clarity in Methodology: While the methodology is generally well-described, certain aspects could be clarified to improve replicability. For example, the specific criteria for sample selection should be detailed further, and the rationale for choosing particular statistical methods should be elaborated.

Discussion of Limitations: The paper would benefit from a more detailed discussion of its limitations. Acknowledging the limitations of the study, such as potential biases in data collection or analysis, would add depth to the discussion and provide a more balanced interpretation of the findings.

Figures and Data Presentation: The figures are generally clear, but some could be enhanced with more detailed captions. Additionally, providing a brief explanation of how the data in the tables were derived would be helpful for readers.

Literature Integration: The discussion section could be strengthened by integrating the findings more closely with existing literature. Highlighting how this study aligns with, contradicts, or expands upon previous research would add value to the interpretation of the results.

Ethical Considerations: While the study adheres to ethical standards, a more explicit statement regarding how ethical guidelines were followed during the research process (e.g., informed consent, data privacy) would be beneficial for transparency.

---

## Round 0.2 · accepted · Accept

The reviewers are happy with the current version, congratulations on the acceptance of your manuscript.